# Lack of elevated pre-ART elastase-ANCA levels in patients developing TB-IRIS

Odin Goovaerts[1,2]*, Marguerite Massinga-Loembé[1,3,4], Pascale Ondoa[5,6],
Ann Ceulemans[1], William Worodria[7,8,9], Harriet Mayanja-Kizza[7,8], Robert Colebunders[10],
Luc Kestens[1], the TB-IRIS Study Group¶

1 Department of Biomedical Sciences, Institute of Tropical Medicine, Antwerp, Belgium, 2 Department of Clinical Sciences, Institute of Tropical Medicine, Antwerp, Belgium, 3 Centre de Recherches Médicales de Lambaréné (CERMEL), Albert Schweitzer Hospital, Lambarene, Gabon, 4 Institut für Tropenmedizin, Universität Tübingen, Tübingen, Germany, 5 African Society for Laboratory Medicine, Addis Ababa, Ethiopia, 6 Amsterdam Institute of Global Health and Development, Academic Medical Center, Department of Global Health, University of Amsterdam, Amsterdam, The Netherlands, 7 Department of Medicine, Mulago Hospital, Kampala, Uganda, 8 Infectious Diseases Institute, Makerere University College of Health Sciences, Kampala, Uganda, 9 Infectious Diseases Network for Treatment and Research in Africa (INTERACT), Kampala, Uganda, 10 Global Health Institute, University of Antwerp, Antwerp, Belgium

¶ The complete membership of the author group can be found in the Acknowledgments.
* ogoovaerts@itg.be

**Data Availability Statement:** All relevant data are within the manuscript and its Supporting Information files.

## Abstract

### Background

Tuberculosis-associated immune reconstitution inflammatory syndrome (TB-IRIS) in HIV-TB co-infected patients receiving antiretroviral therapy (ART) has been linked to neutrophil activation. Anti-neutrophil cytoplasmic antibodies (ANCAs) are also associated with neutrophil activation. Since ANCAs are reportedly skewed in TB and HIV infections, we investigated plasma levels of 7 ANCAs in TB-IRIS patients.

### Methods

We retrospectively compared 17 HIV-TB patients who developed TB-IRIS with controls of similar CD4 count, age and gender who did not ($HIV^+TB^+$ n = 17), HIV-infected patients without TB ($HIV^+TB^-$, n = 17) and 10 HIV-negative ($HIV^-TB^-$) controls. Frozen plasma was collected before ART, at 3 and 9 months of ART, and examined by ELISA for levels of 7 ANCAs directed against; Proteinase 3 (PR3), Myeloperoxidase (MPO), Permeability-increasing protein (BPI), Elastase, Cathepsin, Lysozyme, and Lactoferrin.

### Results

Compared to $HIV^+TB^+$ controls, pre-ART anti-elastase levels were lower in TB-IRIS patients (p = 0.026) and $HIV^-TB^-$ controls (p = 0.044), whereas other ANCAs did not show significant differences between groups at any time point. A significant decrease over time could be observed in TB-IRIS patients during ART for anti -PR3 (p = 0.027), -lysozyme (p = 0.011), and -lactoferrin (p = 0.019). Conversely, $HIV^+TB^+$ controls showed a significant decrease over time for anti -MPO (p = 0.002), -lyzosyme (p = 0.002) and -elastase (p < 0.001).

**Funding:** This study was supported by EC FP6 Specific Targeted Research Project (STREP) grant LSHP-CT-2007-037659-TBIRIS. The funders had no role in study design, data collection and analysis, decision to publish, or preparation of the manuscript.

**Competing interests:** The authors have declared that no competing interests exist.

## Conclusion

The lack of elevated anti-elastase levels in TB-IRIS patients as opposed to HIV⁺TB⁺ controls correspond to previous findings of lowered immune capacity in patients that will develop TB-IRIS. This may suggest a specific role for anti-elastase, elastase or even matrix-metalloproteinases in TB-IRIS. The precise dynamics of neutrophil activation in HIV-TB merits further investigation and could provide more insight in the early mechanisms leading up to TB-IRIS.

## Introduction

A subgroup of HIV-infected patients with a tuberculosis (TB) co-infection who start successful antiretroviral therapy (ART) are at risk of developing a complication called paradoxical TB-associated immune reconstitution inflammatory syndrome (TB-IRIS), despite responding well to preceding TB-treatment [1, 2]. TB-IRIS typically develops within the first 2 months after starting ART, with the majority of cases occurring before 1 month when CD4+ T-cells are being replenished [3, 4]. Though underlying tissue destructive inflammation is heavily associated with TB-IRIS, the immune-pathogenesis is still not well understood [5]. Patients often require additional therapy and/or hospitalization, presenting with a wide array of symptoms. This heterogeneity provides a challenge to distinguish TB-IRIS from other complications, since diagnosis of TB-IRIS still mainly relies on clinical examinations. Though a decade of research is steadily uncovering the mechanisms behind TB-IRIS, there is still a need for reliable laboratory markers to accurately predict patients at risk of the syndrome [6].

Early studies described a high TB-antigen burden, a short interval between TB treatment and ART, and a low CD4+ T cell count prior to ART initiation as the major risk factors of TB-IRIS [7–9]. However, not all HIV-TB patients with advanced disease develop TB-IRIS. Therefore, additional mechanisms such as an atypical restoration of immune responses to TB, or an uncoupling of adaptive and innate immune responses, have been proposed as underlying processes leading up to TB-IRIS [3, 10–12]. Whereas an early role of innate immune cells in the inflammatory cascade during TB-IRIS has become increasingly apparent [13–18], it still remains unclear which pre-ART factors prime the immune system for the inflammatory burst during ART. With that in mind, current theories seem to flow together to form a more complex model of TB-IRIS pathogenesis that spans different aspects of the immune system [19].

Anomalies in the innate arm of the immune system have been previously reported by different studies. The IRIS-typical cytokine storm seems to be dominated by innate factors such as IL-6, which suggests the involvement of monocytes [14, 20–24]. We previously observed lower levels of IL-6 and LPS-binding protein in TB-IRIS patients prior to starting ART, suggesting less innate immune activation during the TB-treatment–ART interval [14]. Conversely, neutrophils and related products such as matrix-metalloproteinases have been closely associated with the tissue-damage observed during ART in TB-IRIS patients [25, 26]. In fact, plasma concentrations of human neutrophil peptide 1–3 and elastase were found to be elevated during TB-IRIS events [27]. Therefore, investigating downstream factors involved in monocyte/neutrophil activation and tissue damage could provide additional markers, which could help elucidate the pathogenesis of TB-IRIS.

Taking a page out of the book of auto-immune diseases, anti-neutrophil cytoplasmic antibodies (ANCAs, a group of antibodies directed against different neutrophilic components), have been widely linked to neutrophil activation and tissue damage in small vessel vasculitides [28, 29]. While anti-Myeloperoxidase (MPO) and anti-Proteinase 3 (PR3) are considered the

major components of the ANCA group, antibodies against other components such as bacterial permeability-increasing protein (BPI), elastase, cathepsin, lysozyme, and lactoferrin have all been described in inflammatory conditions which include; inflammatory rheumatic diseases, inflammatory bowel disease, etc. [30–33]. Interestingly, aberrant ANCA plasma levels have also been associated with chronic infections such as HIV and TB, perhaps triggered by molecular mimicry or persistent host responses to the infection [34–36]. Nonetheless, ANCA findings in HIV and TB remain inconsistent. The objective in this study was therefore to explore ANCA levels in TB-IRIS patients and controls. We hypothesized that pre-ART ANCA levels would be lower in TB-IRIS patients, mirroring sub-optimal pre-ART neutrophil activation, and consequently a diminished ability in these patients to clear the pre-ART antigen load. Nested as a case-control study within a large prospective cohort of TB-IRIS patients, we describe lower pre-ART plasma levels of anti-elastase, but not other ANCAs, in TB-IRIS patients compared to HIV-TB controls.

## Methods

### Study population

Patients in our study were recruited as part of a prospective observational study on the clinical spectrum of HIV-TB and TB-IRIS between 2007 and 2012 at Mulago Hospital, Kampala, Uganda [7, 37, 38]. Enrollment included a cohort of HIV-TB co-infected adults (HIV$^+$TB$^+$) who had been treated for active TB infection for less than 2 months, and a cohort of HIV-infected patients without clinical signs of TB co-infection (HIV$^+$TB$^-$). Recruitment of HIV$^+$TB$^+$ and HIV$^+$TB$^-$ patients occurred in parallel from patients attending Mulago Hospital and patients from each group received a non-nucleoside reverse transcriptase inhibitor-based ART according to the same schedule. An additional group of HIV-negative controls without active TB (HIV$^-$TB$^-$) was also recruited. Patients were excluded in case of prior use of ART, pregnancy, or Grade 3 liver or renal abnormalities. According to Ugandan national guidelines, The median interval between starting TB-treatment and ART was 6 weeks for the whole cohort, including adherence preparation for HIV$^+$TB$^+$ patients. HIV-infected patients were followed up for 10 months in order to monitor paradoxical TB-IRIS development. Sixty (24%) out of 254 HIV-TB co-infected patients developed IRIS (TB-IRIS), whereas HIV$^+$TB$^+$ patients who did not develop IRIS-related symptoms served as non-IRIS controls. In this study, a subset of patients with available plasma samples were semi-randomly selected from each patient-group. To do so, equal numbers of patients from each group were randomly selected from pre-determined clusters of patients with similar CD4 counts ($>$ or $\leq$ 45 cells/µl) and gender. HIV-infected patients had blood samples taken before initiation of ART, at 3 months and at 9 months after starting ART, while HIV-negative controls received no ART and thus had samples taken only once.

### Definitions

*Mycobacterium tuberculosis* infection was diagnosed according to the TB/HIV WHO guidelines [39]. Clinical tests to confirm TB infection included: clinical examination, chest X-rays, abdominal ultrasounds, sputum smear microscopy for acid-fast bacilli, and mycobacterial culture of sputum, aspirate or effusion if available. All suspected TB-IRIS cases were evaluated by the study physicians according to the International Network for the Study of HIV-associated IRIS (INSHI) clinical case-definition, and subsequently classified as TB-IRIS cases by a committee of two co-authors (RC and WW) [1]. TB-IRIS was diagnosed when patients presented with at least 1 major criterion (e.g. enlarged lymph nodes) or 2 minor criteria (e.g. fever and cough) and treatment failure was excluded by monitoring viral loads and CD4 counts.

## Plasma ANCA analysis

Venous blood was drawn into EDTA tubes and plasma was separated from cells by centrifugation and stored at -80˚C. Plasma samples were subsequently thawed and diluted 1:101. Plasma concentrations of 7 ANCA's (PR3, MPO, BPI, Elastase, Cathepsin, Lysozyme, Lactoferrin) were determined by ELISA according to the manufacturer's instructions (Kit: AESKULISA ANCA-Pro; Catalog Number: 3301; Manufacturer: AESKU.DIAGNOSTICS GmbH & Co., Wendelsheim, Germany).

## Ethics statement

The study was approved by: The Research Committee of the Infectious Diseases Institute (IDI), the ethical review board of Makerere University (IRB-Makerere-05_2007), the Uganda National Council of Science and Technology, the institutional review board of the Institute of Tropical Medicine of Antwerp and the Ethics Committees of the Faculties of Medicine of the University of Antwerp (CME_UZA_7/29/157). Written informed consent was obtained from all study participants.

## Statistical analysis

SPSS software (version 17.0) and GraphPad Prism (version 7) was used for statistical analysis, with significance level set at $p < 0.05$. Kruskal-Wallis tests were used to analyse differences between patient groups. A Mann Whitney U test was used to analyse differences in CRP levels between TB-IRIS patients and $HIV^+TB^+$ patients and a Fisher's exact test was used for differences in gender between groups. Changes over time were analysed for each group using Friedman tests (n = 11 in each group, since complete follow up samples are needed for longitudinal analysis). Dunn's multiple comparison post-hoc tests and multiplicity adjusted p-values were used to indicate differences between specific groups and time points.

# Results

## Study population

Nested within a prospective cohort study for TB-IRIS, we compared 17 TB-IRIS patients to equal numbers of $HIV^+TB^+$ and $HIV^+TB^-$ controls. The median (interquartile range (IQR)) number of days between starting ART and TB-IRIS diagnosis was 14 (11–27) days. None of the HIV-infected groups showed significant differences in clinical pre-ART characteristics, except for $HIV^+TB^-$ controls who showed lower CRP levels compared to TB-IRIS patients and $HIV^+TB^+$ controls (p <0.001, Table 1). TB-IRIS patients with available HIV viral loads (n = 11) showed a significant decline in viral load after 2 weeks of ART [5,4 (5,2–5,7) vs. 3,4 (3,2–3,4) Log copies/ml, p = 0.001 using Wilcoxon signed-rank test]. In addition, 10 $HIV^-TB^-$ controls were selected. $HIV^-TB^-$ controls did not differ in age or sex from any of the HIV-infected groups, but had lower CRP levels compared to TB-IRIS patients and $HIV^+TB^+$ controls (p ≤ 0.001).

## Anca levels in TB-IRIS patients and controls

Given the association of ANCAs with inflammatory conditions, we explored plasma levels of 7 different ANCAs in TB-IRIS patients and controls at different intervals before and during ART (Fig 1A). Prior to ART, we observed a lack of elevation in anti-elastase levels for TB-IRIS patients. The plasma levels were significantly lower in TB-IRIS patients (p = 0.026) and $HIV^-TB^-$ controls (p = 0.044) compared to $HIV^+TB^+$ controls. Though anti-elastase levels remained low in TB-IRIS patients, this significance was lost at 3 months and 9 months of

**Table 1. Clinical characteristics of the study population.**

| | TB-IRIS | HIV⁺TB⁺ | HIV⁺TB⁻ | HIV⁻TB⁻ | pᵃ |
|---|---|---|---|---|---|
| | (n = 17) | (n = 17) | (n = 17) | (n = 10) | |
| **Baseline Characteristics** | | | | | |
| Male (n) (%) | 8/17 (47) | 8/17 (47) | 8/17 (47) | 5/10 (50) | 1.000ᵇ |
| Age (Years) | 29 (25–39) | 30 (25–37) | 35 (30–38) | 34 (30–40) | 0.410 |
| CRP (mg/L) | 7,6 (5,8–42,0)ᵉ | 13,0 (4,8–39,0) | 1,3 (0,7–4,4) | 0,9 (0,6–1,6) | <0.001ᶜ |
| # CD4 (cells/µl) | 38 (9,5–134) | 32 (14–123) | 34 (13–133) | - | 0.974 |
| TB-treatment–ART intervalᵈ | 44 (26–70) | 58 (36–65)ᶠ | - | - | 0.634ᴸ |
| Pre-ART—ART initiation (#days) | 16 (0–26) | 2 (0–30)ʰ | 3 (0–7) | - | 0.108 |
| Temperature (˚C) | 36 (36–37) | 36 (36–37) | 36 (36–37)ᵍ | - | 0.435 |
| **Characteristics after 6 months of ART** | | | | | |
| # CD4 (cells/µl) | 173 (95–329)ⁱ | 240 (123–356)ʲ | 214 (136–314)ᵏ | - | 0.666 |
| Temperature (˚C) | 36 (35–36)ⁱ | 36 (36–37)ʲ | 36 (36–36)ᵏ | - | 0.174 |

Data are represented as median and interquartile range unless stated otherwise.

ᵃKruskal-Wallis tests were used to compare groups unless stated otherwise.

ᵇFisher's exact test

ᶜDifferences between specific groups were calculated using Dunn's post-hoc test with multiplicity adjusted p-values: TB-IRIS vs. HIV⁺TB⁺ p>0.999; TB-IRIS vs. HIV⁻TB⁻ **p = 0.005**; TB-IRIS vs. HIV⁻TB⁻ **p<0.001**; HIV⁺TB⁺ vs. HIV⁻TB⁻ **p = 0.009**; HIV⁺TB⁺ vs. HIV⁻TB⁻ **p = 0.001**; HIV⁺TB⁻ vs. HIV⁻TB⁻ p>0.999.

ᵈ#days between initiation of TB-treatment and ART.

ᵉn = 13.

ᶠn = 15.

ᵍn = 15.

ʰn = 16.

ⁱn = 16.

ʲn = 14.

ᵏn = 15.

ᴸMann-Whitney U test. The level of significance was set to P > 0.05 for all tests. CRP = C-Reactive Protein. Note: due to the nature of the study-design, clinical characteristics during follow-up were documented at month 6, rather than months 3 and 9.

ART, since HIV⁺TB⁺ controls showed a significant decrease in anti-elastase levels over time (p < 0.001; Fig 1B). Dunn's post-hoc test revealed a decrease from month 3 to month 9 (p = 0.009) and pre-ART to month 9 (p = 0.002).

In contrast, no significant differences could be observed between groups at any time point for anti- PR3, -MPO, -BPI, -elastase, -cathepsin, -lysozyme, and -lactoferrin (Fig 2). However, a significant decrease over time could be observed in TB-IRIS patients for anti -PR3 (p = 0.027), -lysozyme (p = 0.011), and -lactoferrin (p = 0.019; Fig 3A, 3E and 3F). Dunn's post-hoc test showed a main decrease between pre-ART and the month 9 time point (p = 0.032, p = 0.017, and p = 0.017, respectively). HIV⁺TB⁺ controls showed a significant decrease over time for anti-lyzosyme as well, in addition to a decrease in anti-MPO (p = 0.002 for both; Fig 3B and 3E). For anti-lyzosyme, the decrease was located between the pre-ART and month 9 time point (p = 0.004). For anti-MPO, this decrease was located between month 3 and month 9 (p = 0.032), as well as between pre-ART and month 9 (p = 0.004). Next, we determined the absolute delta-values of changes over time in for anti -PR3, -MPO, -elastase, -lysozyme, and -lactoferrin by subtracting month 9 values from pre-ART values for each patient (Table 2). However, no significant differences could be observed between groups when comparing these values.

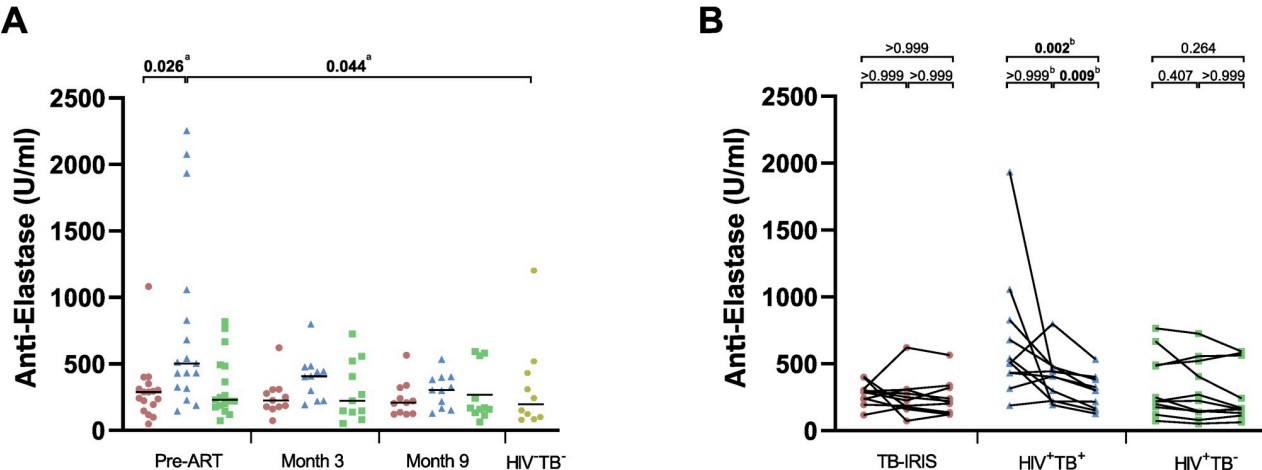

**Fig 1. Plasma levels of anti-elastase in TB-IRIS patients and controls.** Graphs show a comparison between groups (A), as well as changes over time for individual patients (B) in the median plasma levels of anti-elastase (U/ml) measured prior to ART, and at 3 months and 9 months of ART. Significant overall differences between TB-IRIS patients (red circles), HIV+TB+ (blue triangles), HIV+TB- (green squares), and HIV-TB- (yellow hexagons) controls were calculated using Kruskal-Wallis tests, with Dunn's post-hoc test shown as capped lines to highlight specific differences between individual groups (A). Significant variation over time was calculated for each group using Friedman tests, with Dunn's post-hoc test shown as capped lines to highlight specific differences between individual time points (B). Out of 17 TB-IRIS patients with samples available pre-ART, only 11 had samples available at both month 3 and month 9. The total number of patients analysed over time was thus 11 for each group. Analysis were performed between all groups at each time point, and between every time points for each group with the level of significance was set to $P < 0.05$. Non-significant p-values have been omitted from the graphs. Overall Kruskal-Wallis p-value was [a]p = 0.009 and overall Friedman p-value was [b]p ≤ 0.001.

## Discussion

The important role of monocyte and neutrophil reactions in the inflammatory cascade that characterizes TB-IRIS has become a topic of increasing interest for TB-IRIS studies [14, 15, 17, 18] [27, 40]. However, it still remains unclear which pre-ART factors predispose the immune-system to trigger this excessive inflammatory response upon ART initiation. Studies on vasculitis have shown that neutrophils express auto-antigens when exposed to antigens [28]. Subsequent binding of so called anti-neutrophil cytoplasmatic antibodies, or ANCAs, leads to the activation of neutrophils and the release of reactive oxygen species, inflammatory cytokines and the corresponding tissue damage. Taking into account the possible roles of ANCAs in infectious diseases such as TB and HIV, as well as the apparent role of neutrophils in TB-IRIS, measuring ANCA-levels could provide more insight in the activity of neutrophils in TB-IRIS patients. We therefore aimed to explore plasma-levels of 7 different ANCAs in TB-IRIS patients prior to ART, and after 3 and 9 months of follow-up. Since we previously observed lower pre-ART levels of IL-6 and acute phase protein (lipopolysaccharide-binding protein) in TB-IRIS patients, we hypothesized that ANCA levels would also be lower in TB-IRIS patients, mirroring sub-optimal neutrophil activation, and consequently a diminished ability in these patients to clear the pre-ART antigen load.

We thus investigated plasma levels of anti-PR3, -MPO, -BPI, -elastase, -cathepsin, -lyso-zyme, and -lactoferrin in TB-IRIS patients and controls prior to starting ART. Samples taken during 3 and 9 months of ART were also analysed in order to better understand the dynamics of these ANCA's during treatment. Contrary to our hypothesis, most ANCAs that were measured did not show significant direct differences between groups. However, anti-elastase did show lower pre-ART levels in TB-IRIS patients, in line with our theory. In addition, while both TB-IRIS patients and HIV+TB+ controls showed decreasing levels of anti-lysozyme during follow-up, TB-IRIS patients showed a stronger decrease in anti-lactoferrin, whereas HIV+TB+ controls showed a decrease in anti-MPO and -elastase. This finding could suggest

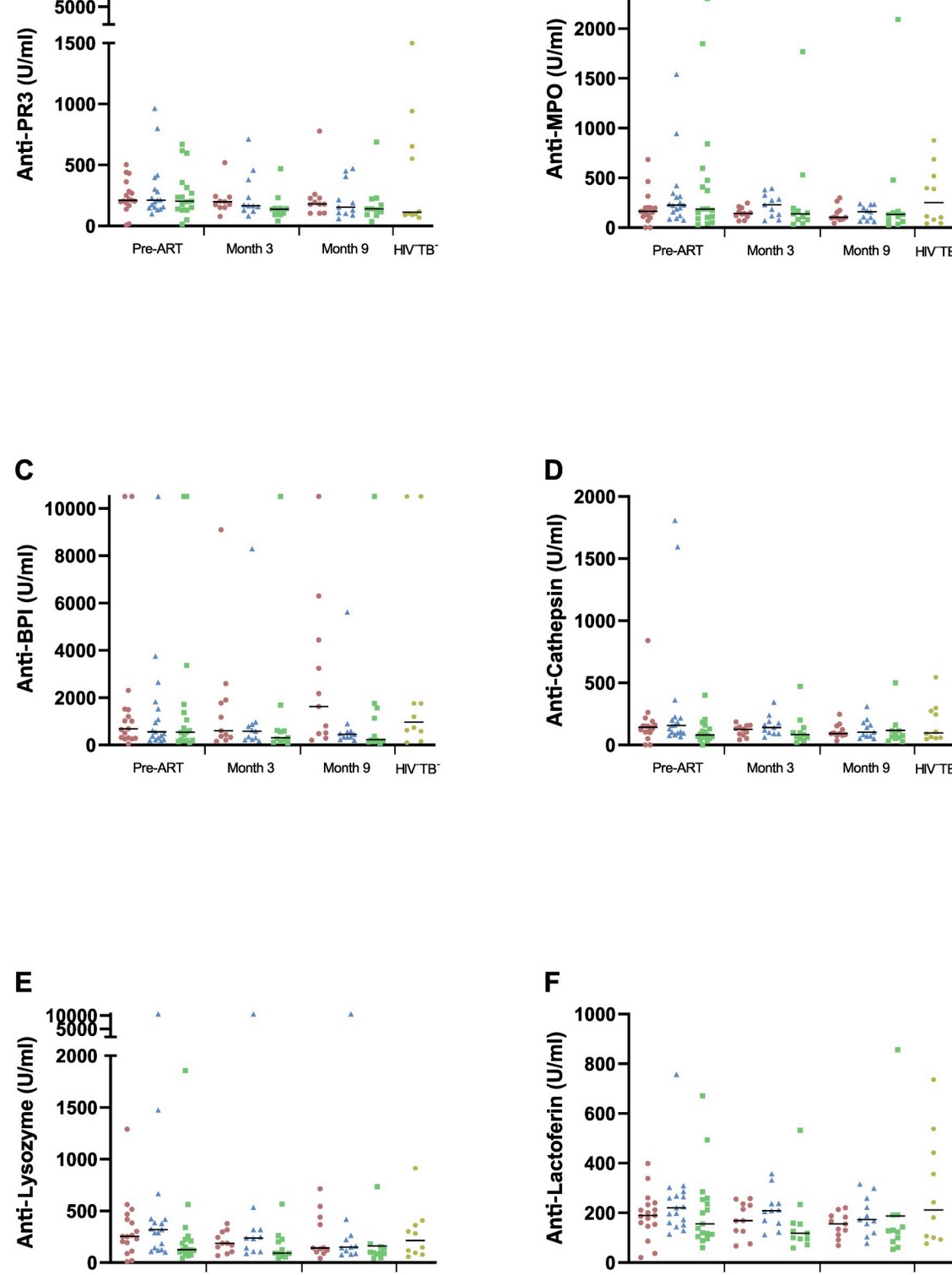

**Fig 2. Plasma levels of other ANCAs in TB-IRIS patients and controls.** Graphs show a comparison between TB-IRIS patients (red circles), HIV+TB+ (blue triangles), HIV+TB- (green squares), and HIV-TB- (yellow hexagons) in the median plasma levels of anti-PR3 (A), -MPO (B), -BPI (C), -cathepsin (D), -lysozyme (E), and -lactoferin (F) (U/ml) measured prior to ART, and at 3 months and

9 months of ART. Analysis were performed between all groups at each time point using Kruskal-Wallis tests, with Dunn's post-hoc test, and the level of significance set to $P < 0.05$. Non-significant p-values have been omitted from the graphs, as no significant values were observed.

diverging ANCA dynamics in TB-IRIS patients. Nonetheless, no significant differences were observed when comparing absolute changes over time (calculated through delta-values). Further research is therefore required to fully understand the ANCA dynamics spanning the TB-IRIS event. Interestingly, an increase in elastase levels has previously been described during ongoing TB-IRIS [27], though no differences were observed prior to ART. While not completely in line with nor contradicting our findings, this does suggest a more prominent role for elastase and its corresponding ANCA in TB-IRIS development. Neutrophilic elastase is associated with tissue remodelling/damage through the activation of matrix-metalloproteinases, which in and of themselves have been linked to TB-IRIS as well [26, 41]. On one hand, one could speculate that the lower pre-ART anti-elastase levels observed here, as compared to HIV+TB+ patients, indicates a higher level of elastase being produced that is consequently binding with these ANCAs. However, increased rather than decreased ANCA levels are far more commonly associated with pathology seen in diseases such as inflammatory bowel disease, linked to increased neutrophil activation [42]. It is therefore more likely that the decreased ANCA levels reflect a decreased level of neutrophil activation in our patients. In fact, HIV$^+$TB$^+$ patients in our study who did not develop complications after starting ART, showed higher ANCA levels compared to not only TB-IRIS patients, but HIV$^-$TB$^-$ controls as well. This suggests that elevated ANCA levels could be a typical response in HIV-TB coinfection, in contrast to HIV$^+$TB$^+$ patients who developed TB-IRIS during ART. Interestingly, our previous observations show a downregulation of immune factors prior to ART, suggesting a lowered capacity of the immune system in TB-IRIS patients to clear the mycobacterial load before ART is initiated [14, 43, 44]. Since the lower pre-ART ANCA levels observed here are in line with our earlier results, a corresponding decrease in neutrophil activation (relative to a more typical HIV$^+$TB$^+$ coinfection), does seem plausible. Since these TB-IRIS patients were recruited under similar clinical conditions as HIV$^+$TB$^+$ controls (i.e. similar CD4 counts and treatment interval), one could argue that a diminished activation of phagocytotic cells such as neutrophils and macrophages would slow down the clearance of TB-bacilli and their antigens in the window between TB-treatment and ART. This fits with the already established theory that a pre-ART reduction in immune capacity could allow a build-up of antigens, which then primes the immune system to over-react when ART is initiated [3]. Once ART is initiated, the corresponding increase in T-helper cell-derived interferon-gamma and the higher antigen-load would then lead to a pathological activation of neutrophils. Nonetheless, while lower anti-elastase levels could thus be an indication of this "immune-paralysis" prior to ART, it remains unclear why other ANCAs did not share this pattern.

As ours was a retrospective study on a condition with unpredictable timing, we could not include plasma taken during TB-IRIS event. We therefore cannot provide any insight in the evolution of ANCA levels during the ongoing TB-IRIS inflammation. Nonetheless, our selected samples span a timeframe both before and after TB-IRIS development, allowing an overview of the events surrounding TB-IRIS development. Moreover, our main finding was observed prior to starting ART, and is therefore not influenced by not including samples taken during the TB-IRIS event.

In conclusion, our study observed a lack of elevated anti-elastase antibodies in TB-IRIS patients, compared to HIV$^+$TB$^+$ controls before starting ART. Nonetheless, none of the other ANCAs were significantly different between groups. Our findings correspond to previous

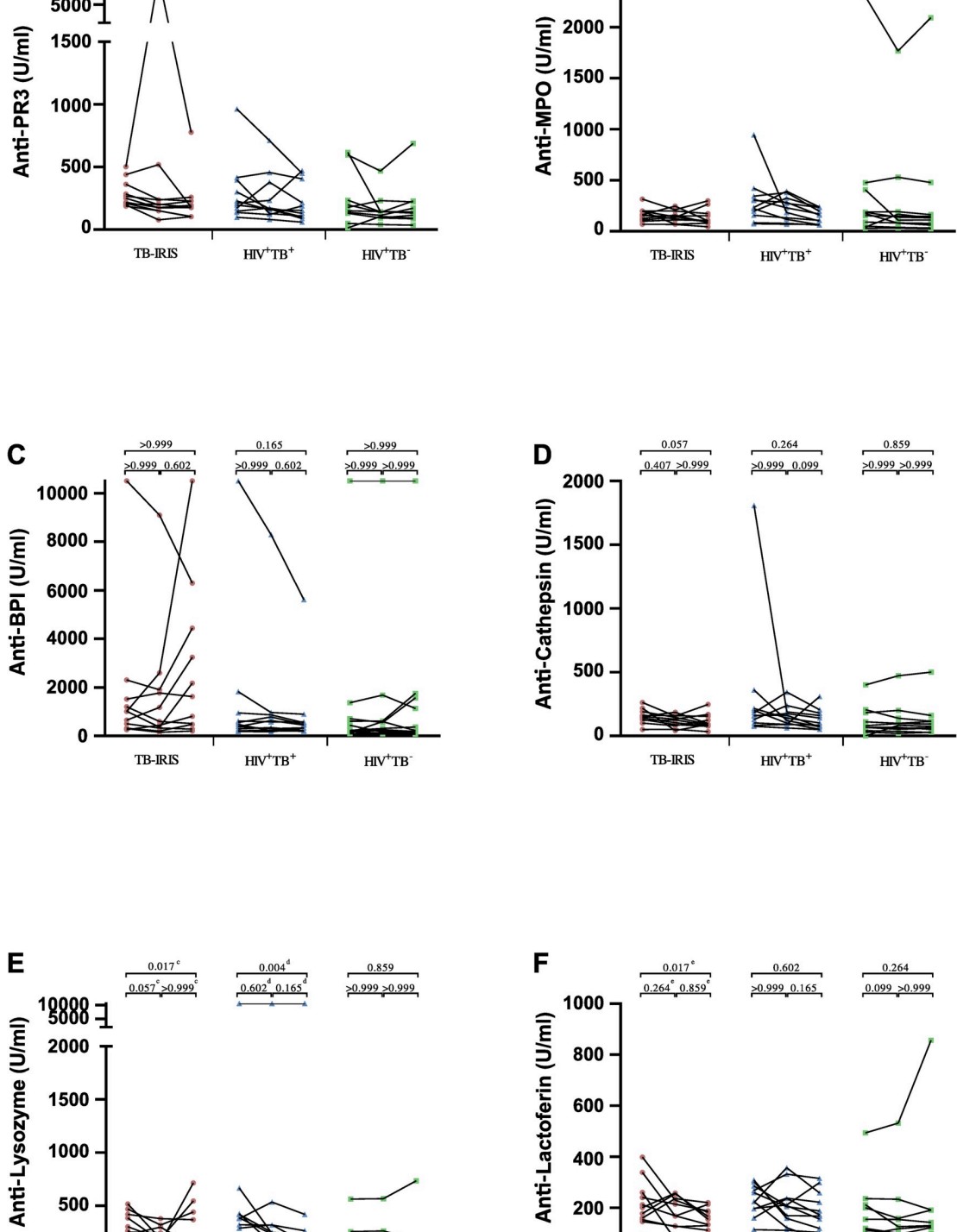

**Fig 3. Plasma levels of other ANCAs over time in TB-IRIS patients and controls.** Graphs show changes over time for individual patients in the median plasma levels of anti- PR3 (A), -MPO (B), -BPI (C), -cathepsin (D), -lysozyme (E), and -lactoferrin (F) (U/ ml) measured prior to ART, and at 3 months and 9 months of ART. Significant variation over time in TB-IRIS patients (red circles), HIV+TB+ (blue triangles), and HIV+TB- (green squares) controls was calculated for each group using Friedman tests, with

Dunn's post-hoc test shown as capped lines to highlight specific differences between individual time points. Out of 17 TB-IRIS patients with samples available pre-ART, only 11 had samples available at both month 3 and month 9. The total number of patients analysed over time was thus 11 for each group. No variation over time was calculated for HIV⁻TB⁻ controls, since only 1 time point was available. Analysis were performed between all groups at each time point, and between every time points for each group with the level of significance was set to P < 0.05. Overall Friedman p-values were [a]p = 0.027, [b]p = 0.002, [c]p = 0.011, [d]p = 0.002, [e]p = 0.019.

**Table 2. Change in ANCA-levels at baseline vs 9 months in absolute (delta-)values.**

|  | TB-IRIS | HIV⁺TB⁺ | HIV⁺TB⁻ | p[a] |
|---|---|---|---|---|
|  | (n = 11) | (n = 11) | (n = 11) |  |
| Anti-PR3 | 73 (24–100) | 38 (-16–153) | 11 (-39–29) | 0.246 |
| Anti-MPO | 25 (8–63) | 143 (15–153) | 20 (6–83) | 0.358 |
| Anti-elastase | 49 (-21–102) | 296 (35–414) | 30 (9–87) | 0.071 |
| Anti-lysozyme | 52 (18–142) | 55 (32–211) | 17 (-15–49) | 0.121 |
| Anti-Lactoferrin | 46 (20–101) | 11 (-32–133) | 12 (-6–31) | 0.225 |

Data are represented as median and interquartile range in U/mL. Values were calculated by subtracting values obtained at month 9 from those obtained at the pre-ART time point (with negative values indicating an increase over time, rather than a decrease).

[a]Kruskal-Wallis tests were used to compare groups. The level of significance was set to P > 0.05 for all tests.

findings of diminished immune-activity in TB-IRIS patients prior to starting ART, and could underscore a specific role for anti-elastase, elastase or even matrix-metalloproteinases in TB-IRIS. The precise dynamics of neutrophil activation in HIV-TB patients prior to starting ART merits further investigated and could provide more insight in the early mechanisms leading up to TB-IRIS.

## Supporting information

**S1 Data.**
(XLSX)

## Acknowledgments

The authors thank the study participants and the study team: D. Mazakpwe, K. Luzinda, P. Lwanga, M. Nakuya, C.O Namujju, C. Ahimbisibwe, J. Namaganda, A. Andama, E. Bazze and H. Kisembo. We thank N. Pakker and the data staff of the Infectious Diseases Network for Treatment and Research in Africa (INTERACT) for assistance with data monitoring and management.

**Members of the TB-IRIS study group:** Institute of Tropical Medicine, Antwerp, Belgium: Luc Kestens, Robert Colebunders, Marguerite Massinga Loembé; Infectious Disease Institute, Kampala, Uganda: Harriet Mayanja, William Worodria; Joint Clinical Research Centre: Harriet Mayanja; Université Libre de Bruxelles, Belgium: Francoise Mascart; VIB, Brussels, Belgium and Vrije Universiteit Brussel, Brussels, Belgium: Rafael van den Bergh; Institut Pasteur de Lille, France: Camille Locht; Academic Medical Centre, Department of Global Health and Amsterdam Institute for Global Health and Development, Amsterdam, The Netherlands: Peter Reiss, Frank Cobelens, Pascale Ondoa, Nadine Pakker; INTERACT, Kampala, Uganda: Roy Mugerwa, Harriet Mayanja, Nadine Pakker, William Worodria.

## Author Contributions

**Conceptualization:** Odin Goovaerts.

**Data curation:** Odin Goovaerts, Ann Ceulemans.

**Formal analysis:** Odin Goovaerts.

**Funding acquisition:** Pascale Ondoa, Robert Colebunders, Luc Kestens.

**Investigation:** Marguerite Massinga-Loembé, Pascale Ondoa, William Worodria, Harriet Mayanja-Kizza, Robert Colebunders.

**Methodology:** Odin Goovaerts, Ann Ceulemans, Luc Kestens.

**Project administration:** Marguerite Massinga-Loembé, Pascale Ondoa, William Worodria, Harriet Mayanja-Kizza, Robert Colebunders, Luc Kestens.

**Resources:** Luc Kestens.

**Supervision:** Odin Goovaerts.

**Validation:** Odin Goovaerts, Ann Ceulemans.

**Visualization:** Odin Goovaerts.

**Writing – original draft:** Odin Goovaerts.

**Writing – review & editing:** Odin Goovaerts.

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
