## [Decision Letter · Decision Letter 0]

11 Sep 2020

PONE-D-20-09937

Lower pre-ART levels of elastase-ANCA in patients developing TB-IRIS

PLOS ONE

Dear Dr. Goovaerts,

Thank you for submitting your manuscript to PLOS ONE. After careful consideration, we feel that it has merit but does not fully meet PLOS ONE’s publication criteria as it currently stands. Therefore, we invite you to submit a revised version of the manuscript that addresses the points raised during the review process.

As you can see from the reviews below, both reviewers suggest minor revisions to the manuscript to improve the clarity of the presentation and discussion of your findings that I hope you will be able to address. 

We look forward to receiving your revised manuscript.

Kind regards,

Leo Carlin

Academic Editor

PLOS ONE

Journal Requirements:

2. Please add the full name and catalog numbers of ELISA kits used in your experiments to the Methods section of your manuscript.

Reviewers' comments:

Reviewer's Responses to Questions

**Comments to the Author**

1. Is the manuscript technically sound, and do the data support the conclusions?

Reviewer #1: Yes

Reviewer #2: Partly

2. Has the statistical analysis been performed appropriately and rigorously? 

Reviewer #1: Yes

Reviewer #2: Yes

3. Have the authors made all data underlying the findings in their manuscript fully available?

Reviewer #1: Yes

Reviewer #2: Yes

4. Is the manuscript presented in an intelligible fashion and written in standard English?

Reviewer #1: Yes

Reviewer #2: Yes

5. Review Comments to the Author

Reviewer #1: This is a case control study of patients at risk of TB-IRIS who had plasma samples collected previously in a cohort study. Plasma samples were collected from participants prior to ART initiation, at 3 months and 9 months post-ART initiation. Cases (patients who developed TB-IRIS during follow up) are compared to controls (patients who similarly had a recent TB diagnosis and were also starting ART during follow up but did not develop TB-IRIS) and a comparator group of patients with HIV infection, who started ART but did not have a TB diagnosis are also included in the analysis.

The study relates to a disease of importance for which the pathophysiology is not fully understood. The rationale is clear and focuses on an area of current scientific interest, the methods appear sound and the results are well presented. The data reported is novel, showing lower levels of anti-elastase ANCA in TB-IRIS patients prior to ART initiation, a surprising finding. Longitudinal results are also reported for several ANCAs.

Limitations of the data are the relatively small sample size which may have resulted in under-powering. However, the findings are not overstated.

The study report should be supplemented with the following for improved interpretation:

1) Line 122: how the controls were selected. Clearly state in the manuscript the inclusion and exclusion criteria for cases and controls and state whether the selection of controls was random (from those available) or by matching and if so on what variables.

2) Table 1 (or elsewhere in methods): include how many days prior to ART initiation the pre-ART sample was collected, was it immediately prior to ART initiation in all participants and similar in cases and controls?

3) In the discussion (e.g. Line 263) the authors argue that low ANCA levels are the pathological finding and this would benefit from a reference to the physiological role of ANCA. Increased rather than decreased ANCA is more commonly associated with pathology, in other (e.g. autoimmune) disease states.

I suggest also the following minor amendments:

a) Avoid a hyphen between “HIV” and patients as this may be mistakenly interpreted to mean HIV negative.

b) Line 66 typographical: should read “complications”

c) Line 149/150 it is a bit unclear on the use of each test, suggest clarify which test was used in each figure legend.

d) In the figures, pre-fix the y axis labels with “anti-” for clarity

Reviewer #2: The authors present a small retrospective longitudinal analysis of neutrophil associated autoantibodies at pre and post ART initiation in HIV infected individuals with TB. Their stated hypothesis is that ANCA will be lower pre-ART in those who subsequently develop IRIS after ART initiation. The interrogation of ANCA in TB HIV is so far minimally documented, with heterogeneity in results and no clear phenotypic link. Despite limited differences noted between the well-controlled groups, the findings were robustly conducted and represent an important finding in this specific disease context. My only major concern is the phrasing of the presentation of results and title, leading with the finding that IRIS associated with lower anti-Elastase, where actually there was no difference in IRIS compared to HIV+TB- controls and rather it was the non-IRIS group which showed elevation. As such non-IRIS TB-HIV showed a significant decline over time, whilst there was no change in the IRIS group. The changes observed over time also appear very small and should at least be compared in absolute values to demonstrate the size of the reduction and indicate whether they decrease to control levels. IRIS is therefore associated with a lack of elevation.

Minor comments.

• That the rational for a hypothesised lower ANCA concentration pre-ART in IRIS patients is counter to the presented argument that IRIS is associated with activated neutrophil associated tissue damage, thus warranting the investigation of ANCA. Neutrophil associated proteins have been shown to be elevated during IRIS, as stated. It would aid the reader to make a clearer link between how low ANCA pre-ART would contribute to elevated neutrophil activation during IRIS but decreased neutrophil activation pre IRIS (line 263). How would lower ANCA represent diminished pathogen clearance?

• The methods should state clearly how and when the HIV+TB- groups were recruited. Were they also initiating ART at enrolment? Table 1 should indicate how long they have received ART (if they had) and ‘treatment interval’ should specify ‘TB treatment interval’. Additionally, Table 1 should include the HIV-TB- group and indicate these are baseline characteristics. The change at 3 and 9 months in these variables should also be included for all longitudinal groups.

• Line 178 – indicate what treatment this refers to – ART?

• Line 203 – the findings of HIV+TB+ is not converse to that for IRIS as both show declining anti-lysozyme. I suggest rewording.

• Fig.1B, x-axis should have time points added

• Figure legends – State median is plotted.

• Fig. 2 legend- add statistical test used. Reword last sentence, it’s unclear.

• The first paragraph of the discussion is quite repetitive of the introduction and should rather focus more on explaining the context of the results.

6. PLOS authors have the option to publish the peer review history of their article (what does this mean?). If published, this will include your full peer review and any attached files.

Reviewer #1: No

Reviewer #2: No

---

## [Author Response · Author response to Decision Letter 0]

11 Oct 2020

Response to reviewers for the manuscript entitled: “Lower pre-ART levels of elastase-ANCA in patients developing TB-IRIS”, PONE-D-20-09937.

General author response: 

We kindly thank the editor and the reviewers for taking the time to review and provide feedback to our manuscript. We understand that this cannot be taken for granted during the current world-wide situation and would thus like to express our sincere appreciation for the efforts made.

Please ensure that your manuscript meets PLOS ONE's style requirements, including those for file naming. The PLOS ONE style templates can be found at https://journals.plos.org/plosone/s/file?id=wjVg/PLOSOne_formatting_sample_main_body.pdf and https://journals.plos.org/plosone/s/file?id=ba62/PLOSOne_formatting_sample_title_authors_affiliations.pdf

Please add the full name and catalog numbers of ELISA kits used in your experiments to the Methods section of your manuscript.

Author response: As requested we have now added this information. 

On lines 143 – 145 we now state: : “ELISA according to the manufacturer’s instructions (Kit: AESKULISA ANCA-Pro; Catalog Number: 3301; Manufacturer: AESKU.DIAGNOSTICS GmbH & Co., Wendelsheim, Germany).”

 

Author’s response to reviewers

----Reviewer #1----

Comment 1: The study report should be supplemented with the following for improved interpretation:

1) Line 122: how the controls were selected. Clearly state in the manuscript the inclusion and exclusion criteria for cases and controls and state whether the selection of controls was random (from those available) or by matching and if so on what variables.

Author response: We agree this should have been mentioned in the original manuscript. To limit heterogeneity in basic parameters such as gender and CD4 count, control groups were semi-randomly selected from clusters of patients. This method was chosen over direct matching, due to the difficulty of lining up gender, age and CD4 count with a limited number of patients (i.e. with available plasma at the required time-points). This was done according to the following criteria: 

First, patients with no available plasma samples were excluded. Then patients were divided into clusters of patients with CD4 counts above and below (including) 45 cells/µL. These clusters were then further divided into male/female clusters. From the resulting clusters, equal numbers of patients were randomly selected to correspond to the CD4 count and gender balance in the TB-IRIS group. Age was analyzed (table 1), but not included in the selection criteria. 

We adapted the sentence on lines 123-126 to the following: “In this study, a subset of patients with available plasma samples were semi-randomly selected from each patient-group. To do so, equal numbers of patients from each group were randomly selected from pre-determined clusters of patients with similar CD4 counts (> or ≤ 45 cells/µl) and gender.”

2) Table 1 (or elsewhere in methods): include how many days prior to ART initiation the pre-ART sample was collected, was it immediately prior to ART initiation in all participants and similar in cases and controls?

Author response: We thank the reviewer for spotting this oversight, it is indeed essential information to include. We have now added a line in table 1 listing the number of days of sample collection prior to ART initiation (median – interquartile range).

In table 1 on page 7, we now state: 

“Pre-ART - ART initiation (#days) //16 (0 – 26) // 2 (0 – 30)[h] // 3 (0 – 7) // 0.108. [h]n=16.”

For TB-IRIS, HIV+TB+ and HIV+TB- patients, respectively. A Kruskal-Wallis test was used to calculate significance as with other data in the table. For transparency towards the reviewers, the Dunn’s multiple comparison test showed: TB-IRIS vs. HIV+TB+ p = 0.677, TB-IRIS vs HIV+TB- p = 0.108, and HIV+TB+ vs HIV+TB- p >0.999. Please note that one patient in the HIV+TB+ group had missing data and was not included in the calculations.

3) In the discussion (e.g. Line 263) the authors argue that low ANCA levels are the pathological finding and this would benefit from a reference to the physiological role of ANCA. Increased rather than decreased ANCA is more commonly associated with pathology, in other (e.g. autoimmune) disease states.

Author response: The reviewer raises an interesting viewpoint. Indeed, increased ANCA levels are more common in similar pathologies by far. In fact, at first glance one would expect to observe higher ANCA levels in TB-IRIS patients, as it is a tell-tale inflammatory condition. This may still be true when the IRIS event occurs during ART, which is characterized by the inflammation in question (though unfortunately, we could not test this hypothesis). That being said, more and more studies throughout the years have been observing paradoxical signs of “lower immune reactions” prior to ART in these patients (including our previous studies). Leading to the hypothesis that a form of “immune paralysis” could be taking place. Though not strictly pathological (i.e. the patient is not experiencing immediate pathological signs), this would allow a build-up of antigens. This would then turn pathological, once the immune system “wakes up and lets loose”, if you will. 

We appreciate the feedback given by the reviewer in this comment, as along with comments made by reviewer #2, it allowed us to delve deeper into the meaning of our findings. As requested, we have now added a statement in the discussion to emphasize the pathological role of increased, rather than decreased ANCA levels, with a reference. We also made an effort to clarify the exact nature of lower pre-ART ANCA levels in reference to TB-IRIS pathogenesis during ART. 

On lines 281-303 we now state: “On one hand, one could speculate that the lower pre-ART anti-elastase levels observed here, as compared to HIV+TB+ patients, indicates a higher level of elastase being produced that is consequently binding with these ANCAs. However, increased rather than decreased ANCA levels are far more commonly associated with pathology seen in diseases such as inflammatory bowel disease, linked to increased neutrophil activation [42]. It is therefore more likely that the decreased ANCA levels reflect a decreased level of neutrophil activation in our patients. In fact, HIV+TB+ patients in our study who did not develop complications after starting ART, showed higher ANCA levels compared to not only TB-IRIS patients, but HIV-TB- controls as well. This suggests that elevated ANCA levels could be a typical response in HIV-TB coinfection, in contrast to HIV+TB+ patients who developed TB-IRIS during ART.

Interestingly, our previous observations show a downregulation of immune factors prior to ART, suggesting a lowered capacity of the immune system in TB-IRIS patients to clear the mycobacterial load before ART is initiated [14,43,44]. Since the lower pre-ART ANCA levels observed here are in line with our earlier results, a corresponding decrease in neutrophil activation (relative to a more typical HIV+TB+ coinfection), does seem plausible. Since these TB-IRIS patients were recruited under similar clinical conditions as HIV+TB+ controls (i.e. similar CD4 counts and treatment interval), one could argue that a diminished activation of phagocytotic cells such as neutrophils and macrophages would slow down the clearance of TB-bacilli and their antigens in the window between TB-treatment and ART. This fits with the already established theory that a pre-ART reduction in immune capacity could allow a build-up of antigens, which then primes the immune system to over-react when ART is initiated [3]. Once ART is initiated, the corresponding increase in T-helper cell-derived interferon-gamma and the higher antigen-load would then lead to a pathological activation of neutrophils. Nonetheless, while lower anti-elastase levels could thus be an indication of this “immune-paralysis” prior to ART, it remains unclear why other ANCAs did not share this pattern.”

--Minor Comments:I suggest also the following minor amendments:

a) Avoid a hyphen between “HIV” and patients as this may be mistakenly interpreted to mean HIV negative.

Author response: Thank you for the suggestion, we started by replacing every instance of HIV-patient with “HIV patient”. However, upon further discussion by one of our study physicians (Dr. Colebunders), we decided to replace these instances with “HIV-infected patients” to lower any stigmatizing impressions. 

b) Line 66 typographical: should read “complications”

Author response: Adjusted as indicated by the reviewer.

c) Line 149/150 it is a bit unclear on the use of each test, suggest clarify which test was used in each figure legend.

Author response: Upon reflection, we agree that the phrasing here was not very elegant. Most comparisons made in the manuscript were performed with either the Kruskal-Wallis test (group differences) or the Friedman test (change over time). Mann-whitney U tests were only used to calculate differences in CRP levels, as data was only available for two groups. Similarly, Fisher’s exact tests were only used to calculate differences for binominal data, i.e. gender. We have now clarified this in the statistics section and added the precise tests used to figure legend 2, where it was indeed missing. 

We now state on lines 154-157: “Kruskal-Wallis tests were used to analyze differences between patient groups. A Mann Whitney U test was used to analyze differences in CRP levels between TB-IRIS patients and HIV+TB+ patients and a Fisher’s exact test was used for differences in gender between groups.”

We now state on lines 226-228 in figure legend 2: “Analysis were performed between all groups at each time point using Kruskal-Wallis tests, with Dunn’s post-hoc test, and the level of significance set to P < 0.05. Non-significant p-values have been omitted from the graphs, as none were observed.”

d) In the figures, pre-fix the y axis labels with “anti-” for clarity

Author response: Thank you, we appreciate the advice. “Anti-“ has now been added to each y-axis label.

 

----Reviewer #2----

Comment 1: The authors present a small retrospective longitudinal analysis of neutrophil associated autoantibodies at pre and post ART initiation in HIV infected individuals with TB. Their stated hypothesis is that ANCA will be lower pre-ART in those who subsequently develop IRIS after ART initiation. The interrogation of ANCA in TB HIV is so far minimally documented, with heterogeneity in results and no clear phenotypic link. Despite limited differences noted between the well-controlled groups, the findings were robustly conducted and represent an important finding in this specific disease context. My only major concern is the phrasing of the presentation of results and title, leading with the finding that IRIS associated with lower anti-Elastase, where actually there was no difference in IRIS compared to HIV+TB- controls and rather it was the non-IRIS group which showed elevation. As such non-IRIS TB-HIV showed a significant decline over time, whilst there was no change in the IRIS group. The changes observed over time also appear very small and should at least be compared in absolute values to demonstrate the size of the reduction and indicate whether they decrease to control levels. IRIS is therefore associated with a lack of elevation.

Author response: We thank the reviewer for this honest feedback, and we agree with the assessment that our findings are more a “lack of elevation” rather than a “decrease” in anti-elastase levels. While we do believe that the perspective of the narration in our manuscript should be kept on the TB-IRIS side (as this is after all the anomaly), the HIV+TB+ controls are indeed the ones that show elevated levels. We can therefore assume that the anti-elastase levels observed in these patients are what we should expect from an HIV-TB coinfection in this population, and TB-IRIS patients do not seem to reach these elevated levels. As we do not want to overstate our findings, we made an effort to adjust the phrasing in our manuscript to a more accurate depiction of the difference in plasma levels.

We adapted the title to a more reserved statement, now stating: “Lack of elevated pre-ART elastase-ANCA levels in patients developing TB-IRIS”

In the abstract, we now rephrased the conclusion to emphasize the lack of elevated levels. We now state on lines 51-52: “The lack of elevated anti-elastase levels in TB-IRIS patients as opposed to HIV+TB+ controls correspond to previous findings of lowered immune capacity in patients that will develop TB-IRIS.”

In the results section, we now emphasize the lack of elevation, rather than plasma levels just being lower. On lines 187-189, we now state: “Prior to ART, we observed a lack of elevation in anti-elastase levels for TB-IRIS patients. The plasma levels were significantly lower in TB-IRIS patients (p = 0.026) and HIV-TB- controls (p = 0.044) compared to HIV+TB+ controls.”

In the discussion, we now highlight the fact that the HIV+TB+ patients in fact had the higher levels in this population. On lines 286-288, we now state: “In fact, HIV+TB+ patients in our study who did not develop complications after starting ART, showed higher ANCA levels compared to not only TB-IRIS patients, but HIV-TB- controls as well. This suggests that elevated ANCA levels could be a typical response in HIV-TB coinfection, in contrast to HIV+TB+ patients who developed TB-IRIS during ART.”

And in the conclusion, on lines 312-314, we adapted the first sentence to: “In conclusion, our study observed a lack of elevated anti-elastase antibodies in TB-IRIS patients, compared to HIV+TB+ controls before starting ART. “

Finally, we now include a comparison in absolute values in a new table: page 10, table 2. We calculated de delta values by subtracting month 9 from pre-ART values, to indicate the absolute changes over time as requested. However, when performing Kruskal-Wallis tests, no significant differences could be observed in a comparison between groups (, although a trend could still be observed for anti-elastase, p = 0.071). As our main finding remains the lack of elevated anti-elastase levels prior to ART, this does not impact our conclusions in a major way. 

In the results section on lines 218-221, we now describe this finding: “ Next, we determined the absolute delta-values of changes over time in for anti -PR3, -MPO, -elastase, -lysozyme, and -lactoferrin by subtracting month 9 values from pre-ART values for each patient (Table 2). However, no significant differences could be observed between groups when comparing these values.”

Since this data was not present in the manuscript before, we now also include this finding in the discussion by making our statement on changes over time a bit more reserved. We now state on lines 273-275: “Nonetheless, no significant differences could be observed when comparing absolute changes over time (calculated through delta-values). Further research is therefore required to fully understand the ANCA dynamics spanning the TB-IRIS event.” 

--Minor comments.

Comment 2: That the rational for a hypothesised lower ANCA concentration pre-ART in IRIS patients is counter to the presented argument that IRIS is associated with activated neutrophil associated tissue damage, thus warranting the investigation of ANCA. Neutrophil associated proteins have been shown to be elevated during IRIS, as stated. It would aid the reader to make a clearer link between how low ANCA pre-ART would contribute to elevated neutrophil activation during IRIS but decreased neutrophil activation pre IRIS (line 263). How would lower ANCA represent diminished pathogen clearance?

Author response: We follow the reviewer’s thoughts on this as well. It is indeed rather contradictory at first glance and, given the already paradoxical nature of TB-IRIS, we agree that we must make efforts to be as clear to the reader as possible. 

To briefly illustrate the theory: lower-ANCA levels could be linked to lower neutrophil activation. When exposed to similar clinical conditions (CD4 counts, treatment intervals), this could mean a lower phagocytotic capacity and thus a higher amount of antigens that is not cleared before ART is initiated; once ART starts, the ART-associated increase in T-cells and interferon-gamma would then allow the neutrophils to become activated again and now respond to this increased antigen-load in a pathological way. This is based on the theory published by Barber et al. (reference 3, Barber DL, et a. Immune reconstitution inflammatory syndrome: the trouble with immunity when you had none. Nat Rev Microbiol. Nature Publishing Group; 2012;10: 150–6. doi:10.1038/nrmicro2712)

Taking the feedback from reviewer #1 into account as well, we have now rewritten this part of the discussion into a more comprehensive argument. We now touch on the much more common link between increased ANCA levels and pathology/neutrophil activation and discuss the fact that HIV+TB+ patients in fact had the higher ANCA levels in this context (compared to healthy controls as well). We then moved on to more clearly describe how these comparatively lower pre-ART ANCA levels in TB-IRIS, in concert with our previous findings, reflect a diminished phagocytotic capacity. We then linked the resulting increased antigen-load with the innate-priming theory that was published years ago by Barber et al. We also briefly explained what this theory means, so as to provide more clarity to the reader. On lines 283 – 303, we now state:

“However, increased rather than decreased ANCA levels are far more commonly associated with pathology seen in diseases such as inflammatory bowel disease, linked to increased neutrophil activation [42]. It is therefore more likely that the decreased ANCA levels reflect a decreased level of neutrophil activation in our patients. In fact, HIV+TB+ patients in our study who did not develop complications after starting ART, showed higher ANCA levels compared to not only TB-IRIS patients, but HIV-TB- controls as well. This suggests that elevated ANCA levels could be a typical response in HIV-TB coinfection, in contrast to HIV+TB+ patients who developed TB-IRIS during ART.

Interestingly, our previous observations show a downregulation of immune factors prior to ART, suggesting a lowered capacity of the immune system in TB-IRIS patients to clear the mycobacterial load before ART is initiated [14,43,44]. Since the lower pre-ART ANCA levels observed here are in line with our earlier results, a corresponding decrease in neutrophil activation (relative to a more typical HIV+TB+ coinfection), does seem plausible. Since these TB-IRIS patients were recruited under similar clinical conditions as HIV+TB+ controls (i.e. similar CD4 counts and treatment interval), one could argue that a diminished activation of phagocytotic cells such as neutrophils and macrophages would slow down the clearance of TB-bacilli and their antigens in the window between TB-treatment and ART. This fits with the already established theory that a pre-ART reduction in immune capacity could allow a build-up of antigens, which then primes the immune system to over-react when ART is initiated [3]. Once ART is initiated, the corresponding increase in T-helper cell-derived interferon-gamma and the higher antigen-load would then lead to a pathological activation of neutrophils. Nonetheless, while lower anti-elastase levels could thus be an indication of this “immune-paralysis” prior to ART, it remains unclear why other ANCAs did not share this pattern.”

To streamline our hypothesis with the information provided in the discussion, we have now rephrased the hypothesis on page 4 to better explain the link between lower ANCA levels and antigen clearance. On lines 102-104, we now state: “We hypothesized that pre-ART ANCA levels would be lower in TB-IRIS patients, mirroring sub-optimal neutrophil activation and consequently a diminished ability in these patients to clear the pre-ART antigen load.”

Comment 3a: The methods should state clearly how and when the HIV+TB- groups were recruited. Were they also initiating ART at enrolment? 

Author response: We thank the reviewer for helping us to increase the clarity of the methods and table. HIV+TB- patients were in fact recruited as a second cohort within the scope of the larger study. As described in reference 38, the HIV+TB- cohort was designed to detect the occurrence of ART-associated TB and “unmasking” TB-IRIS in a group of HIV patients without symptoms of TB prior to ART. (Worodria W, et al. Antiretroviral treatment-associated tuberculosis in a prospective cohort of HIV-infected patients starting ART. Clin Dev Immunol. 2011;2011: 758350. doi:10.1155/2011/758350). Unfortunately, this cohort was less successful in generating adequate numbers of unmasking TB-IRIS cases and could therefore only serve as a control group for paradoxical TB-IRIS. Both cohorts of patients were recruited at the same time and therefore followed exactly the same ART schedule (sampling during enrollment, closely followed by a ART-initiation “baseline” visit, and subsequent follow-up visits to monitor the development of their health-situation). 

We have now re-arranged and enhanced the study population section to state the parallel nature of both cohorts more clearly, and to emphasize the start of ART in both groups. On lines 114-116, we now state: “Enrollment included a cohort of HIV-TB co-infected adults (HIV+TB+) who had been treated for active TB infection for less than 2 months, and a cohort of HIV patients without clinical signs of TB co-infection (HIV+TB-). Recruitment of HIV+TB+ and HIV+TB- patients occurred in parallel from patients attending Mulago Hospital and patients from each group received a non-nucleoside reverse transcriptase inhibitor-based ART according to the same schedule.”

Comment 3b: Table 1 should indicate how long they have received ART (if they had) and ‘treatment interval’ should specify ‘TB treatment interval’

Author Response: We have now more clearly stated “TB-treatment – ART interval” in the row-header and kept the “d” reference in the footnote, elaborating with “#days between initiation of TB-treatment and ART”.

As also requested by reviewer #1, we have now added a row to table 1 to specify the number of days between the Pre-ART sample and the initiation of ART. This should clarify that HIV+TB- patients started receiving ART around the same relative time as the other patients. On page 7, in table 1, we now state: “Pre-ART - ART initiation (#days) //16 (0 – 26) // 2 (0 – 30)h // 3 (0 – 7) // 0.108. hn = 16”

Comment 3c: Additionally, Table 1 should include the HIV-TB- group and indicate these are baseline characteristics. The change at 3 and 9 months in these variables should also be included for all longitudinal groups.

Author Response: We understand the need to include additional data on these clinical characteristics. As requested, we have now included a column for the HIV-TB- group in table 1 as well. Unfortunately, due to a combination of study-design and missing data, not all clinical characteristics were available for every patient during follow-up. CRP for example was not measured at time-points other than enrolment and can thus not be provided. In addition, although temperature and CD4 counts were measured during follow-up, this was performed at month 6 rather than during months 3 and 9. Nonetheless, we agree with the need to provide as much follow-up data as possible to provide adequate clarity and transparency to the reader. We therefore opted to include this data in table 1. 

In table 1 on page 7, we now provide a header stating, “baseline characteristics” and a header stating, “Characteristics after 6 months of ART”. We included a disclaimer in the footnote as well, stating “Note: due to the nature of the study-design, clinical characteristics during follow-up were documented at month 6, rather than months 3 and 9.” This should provide the reader with more insight in the follow-up characteristics of the patients, while avoiding confusion between the month 6 vs. month 3-9 time points.

The additional data provided in table one is as follows: 

“ # CD4 (cells/µl) // TB-IRIS: 173 (95 - 329)[i] //HIV+TB+: 240 (123 - 356)[j] // HIV+TB-: 214 (136 - 314)[k] // p = 0.666. For transparency, the Dunn’s multiple comparison test showed: TB-IRIS vs. HIV+TB+ p >0.999, TB-IRIS vs HIV+TB- p >0.999, and HIV+TB+ vs HIV+TB- p >0.999. [i]n=16. [j]n=14. [k]n=15”

“Temperature (°C) // TB-IRIS: 36 (35 - 36)[i] //HIV+TB+: 36 (36 - 37) [j] // HIV+TB-: 36 (36 - 36)[k] // P = 0.174. For transparency, the Dunn’s multiple comparison test showed: TB-IRIS vs. HIV+TB+ p >0.999, TB-IRIS vs HIV+TB- p = 0.186, and HIV+TB+ vs HIV+TB- p >0.999. [i]n=16. [j]n=14. [k]n=15”

Please note that a few patients (1, 3, and 2, respectively) in each group had missing data and were not included in the calculations

Comment 4: Line 178 – indicate what treatment this refers to – ART?

Author response: Thank you for pointing this out, we do have 2 treatments in our cohort and this needs to stay clear. We were referring to ART treatment and have now clarified this in the text. 

We now state on lines 186-187: “Given the association of ANCAs with inflammatory conditions, we explored plasma levels of 7 different ANCAs in TB-IRIS patients and controls at different intervals before and during ART.”

Comment 5: Line 203 – the findings of HIV+TB+ is not converse to that for IRIS as both show declining anti-lysozyme. I suggest rewording.

Author response: We have now clarified the sentence by removing the word conversely and specifying the same decrease in anti-lyzosyme.

On lines 214-216, we now state: “HIV+TB+ controls showed a significant decrease over time for anti-lyzosyme as well, in addition to a decrease in anti-MPO (p = 0.002 for both; Fig 3B,E). For anti-lyzozyme, the decrease was located between the pre-ART and month 9 time point (p= 0.004).For anti-MPO, this decrease was located between month 3 and month 9 (p = 0.032), as well as between pre-ART and month 9 (p = 0.004).”

Comment 6: Fig.1B, x-axis should have time points added

Author response: We agree that this needed to be clarified. We have now added the time-points to the x-axis I figure 1B, as well as in figure 3. 

Comment 7: Figure legends – State median is plotted.

Author response: We thank the reviewer for spotting this oversight. We have now added the statement “median plasma levels” to each figure legend. 

Comment 8: Fig. 2 legend- add statistical test used. Reword last sentence, it’s unclear.

Author response: This was indeed unclear. We have now added the name of the test and clarified the last sentence.

We now state in the legend of figure 2 on lines 226-228: “Analysis were performed between all groups at each time point using Kruskal-Wallis tests, with Dunn’s post-hoc test and the level of significance set to P < 0.05. Non-significant p-values have been omitted from the graphs, as no significant values were observed.”

Comment 9: The first paragraph of the discussion is quite repetitive of the introduction and should rather focus more on explaining the context of the results.

Author response: We understand that the first paragraph of the discussion was repetitive of the introduction, as it did contain certain redundancies. We have now condensed the first part of the paragraph, as this did not need to be so elaborate. We’ve also added a sentence to keep the focus of the paragraph on ANCA-levels and neutrophil-activation, and updated the hypothesis at the end to do the same.

On lines 249-251, we now state: “ The important role of monocyte and neutrophil reactions in the inflammatory cascade that characterizes TB-IRIS has become a topic of increasing interest for TB-IRIS studies[14,15,17,18, 27,40]. However, it still remains unclear which pre-ART factors predispose the immune-system to trigger this excessive inflammatory response upon ART initiation.”

And on lines 256-258: “Taking into account the possible roles of ANCAs in infectious diseases such as TB and HIV, as well as the apparent role of neutrophils in TB-IRIS, measuring ANCA-levels could provide more insight in the activity of neutrophils in TB-IRIS patients. We therefore aimed…”

And lines 261-263 :”We hypothesized that ANCA levels would also be lower in TB-IRIS patients, mirroring sub-optimal neutrophil activation, and consequently a diminished ability in these patients to clear the pre-ART antigen load”

---

## [Decision Letter · Decision Letter 1]

17 Dec 2020

Lack of elevated pre-ART elastase-ANCA levels in patients developing TB-IRIS

PONE-D-20-09937R1

Dear Dr. Goovaerts,

We’re pleased to inform you that your manuscript has been judged scientifically suitable for publication and will be formally accepted for publication once it meets all outstanding technical requirements.

Kind regards,

Leo Carlin

Academic Editor

PLOS ONE

Additional Editor Comments (optional):

Thanks for revising your manuscript. As you can see the reviewers were both happy that the revisions had addressed their concerns, but request some minor wording changes that I agree should be made, however, this will not require subsequent external review.

Reviewers' comments:

Reviewer's Responses to Questions

**Comments to the Author**

1. If the authors have adequately addressed your comments raised in a previous round of review and you feel that this manuscript is now acceptable for publication, you may indicate that here to bypass the “Comments to the Author” section, enter your conflict of interest statement in the “Confidential to Editor” section, and submit your "Accept" recommendation.

Reviewer #1: All comments have been addressed

Reviewer #2: (No Response)

2. Is the manuscript technically sound, and do the data support the conclusions?

Reviewer #1: Yes

Reviewer #2: Yes

3. Has the statistical analysis been performed appropriately and rigorously? 

Reviewer #1: Yes

Reviewer #2: Yes

4. Have the authors made all data underlying the findings in their manuscript fully available?

Reviewer #1: Yes

Reviewer #2: Yes

5. Is the manuscript presented in an intelligible fashion and written in standard English?

Reviewer #1: Yes

Reviewer #2: Yes

6. Review Comments to the Author

Reviewer #1: Please correct a typographical error to the headings of Table 1. Two columns currently have identical headings.

Reviewer #2: The authors have addressed all my concerns with this revised manuscript. The narrative is much clear to follow now.

Very minor comments:

Table 1 -

The n=10 column should be HIV-TB- (thank you for including this group)

New row 'Pre-ART - ART initiation (#days)' title is not clear what this means. This can be clarified in the footnotes.

Table 2 - thank you for including this new data

Title 'Change in ANCA-levels over time' is not accurate as the values given are the difference at baseline vs 9 months, change over time would indicate 9 months vs baseline. This may be confusing at first read. I suggest to either change to negative values or reword the title.

The new working on line 104 :mirroring sub-optimal neutrophil activation, may be clearer if changed to "mirroring sub-optimal neutrophil activation pre-ART,"

7. PLOS authors have the option to publish the peer review history of their article (what does this mean?). If published, this will include your full peer review and any attached files.

Reviewer #1: No

Reviewer #2: **Yes: **Anna Coussens

---

## [Editor Report · Acceptance letter]

22 Dec 2020

PONE-D-20-09937R1 

Lack of elevated pre-ART elastase-ANCA levels in patients developing TB-IRIS 

Dear Dr. Goovaerts:

I'm pleased to inform you that your manuscript has been deemed suitable for publication in PLOS ONE. Congratulations! Your manuscript is now with our production department. 

Kind regards, 

on behalf of

Dr. Leo Carlin 

Academic Editor

PLOS ONE